# Recent Advances in Biological Applications of Aptamer-Based Fluorescent Biosensors

**DOI:** 10.3390/molecules28217327

**Published:** 2023-10-29

**Authors:** Minhyuk Lee, Seonhye Shin, Sungjee Kim, Nokyoung Park

**Affiliations:** 1Department of Chemistry, Pohang University of Science and Technology, Pohang 37673, Republic of Korea; lmh92@postech.ac.kr (M.L.); sungjee@postech.ac.kr (S.K.); 2Department of Chemistry, The Natural Science Research Institute, Myongji University, 116 Myongji-ro, Yongin-si 17058, Republic of Korea; ssh8494@naver.com

**Keywords:** aptamer, aptamer-based sensors, biosensor, fluorescence, signal amplification strategies

## Abstract

Aptamers have been spotlighted as promising bio-recognition elements because they can be tailored to specific target molecules, bind to targets with a high affinity and specificity, and are easy to chemically synthesize and introduce functional groups to. In particular, fluorescent aptasensors are widely used in biological applications to diagnose diseases as well as prevent diseases by detecting cancer cells, viruses, and various biomarkers including nucleic acids and proteins as well as biotoxins and bacteria from food because they have the advantages of a high sensitivity, selectivity, rapidity, a simple detection process, and a low price. We introduce screening methods for isolating aptamers with q high specificity and summarize the sequences and affinities of the aptamers in a table. This review focuses on aptamer-based fluorescence detection sensors for biological applications, from fluorescent probes to mechanisms of action and signal amplification strategies.

## 1. Introduction

Abnormal concentrations in various biomolecules, including macromolecules such as nucleic acids, proteins, and carbohydrates and lipid small molecules such as metabolites, perform various functions in the body, can be indicators of disease states, and are therefore considered detection targets as disease biomarkers [1,2,3]. In the biomedical field, the precise and rapid detection of disease biomarkers, viruses, and pathogens is important for the early diagnosis of diseases and the monitoring of health statuses [4,5]. However, the selective detection of trace amounts of biomarkers in complex biological environments such as blood is a factor that makes the diagnosis of diseases difficult [6,7]. To overcome this problem, molecular recognition elements with a high affinity and selectivity for the target biomarker and sensitive detection systems (sensors) were required, and aptamer-based biosensors have been developed to meet these needs. In addition, disease prevention through the detection of pathogenic bacteria, biotoxins, antibiotic residues, etc. present in contaminated food is also one of the main purposes of aptamer biosensors [8]. Previous food analysis methods employing sophisticated instruments are accurate and sensitive, but there are the issues that instruments are bulky, expensive, nonportable, and require skilled technology [9]. Aptamer biosensors are a promising alternative for solving these issues because they are inexpensive and have a simple detection process.

### 1.1. About Aptamers

Aptamers are short, single-stranded RNA or DNA oligonucleotides of 20–100 nucleotides that can specifically recognize and bind to targets with a unique three-dimensional folded structure. Aptamers can specifically recognize targets and bind with a high affinity through noncovalent interactions such as hydrogen bonding, van der Waals forces, electrostatic interactions, hydrophobic effects, and π-π stacking, as well as shape complementarity due to the unique folding structure [10]. Although the nucleic acid forming the aptamer consists of only four nucleobases, it has been found to be enough to obtain a variety of structures similar to antibodies consisting of 20 amino acids, allowing it to have affinity for a wide range of targets [11]. Among various molecular recognition elements, aptamers are receiving great attention as important molecular recognition tools in clinical, biomedical, pharmaceutical, and food safety fields because they have the following advantages compared to other molecular recognition elements, including enzymes and antibodies: (1) high thermal and physiological stability, (2) high affinity for the target, (3) low immunogenicity, (4) low production cost, (5) wide targeting range (metal ions, small organic molecules, nucleic acids, proteins, cells, etc.), (6) easy functional introduction, and (7) small size [12]. Aptamers are mainly screened and selected in vitro through an iterative technique called SELEX (Systematic Evolution of Ligands by Exponential Enrichment).

In 1990, the first RNA aptamer was successfully screened from an RNA library through the SELEX method. The first DNA aptamer was reported in 1992, and most aptamers have since been screened through SELEX [13,14,15,16]. The typical SELEX process consists of three steps: binding, separation, and amplification (Figure 1a). In the binding step, a combinatorial DNA/RNA library containing 10^15^–10^16^ different random sequence regions (20–60 nt) is incubated with the target to be bound. In the separation step, the unbound sequences with the target are separated and removed from sequences with affinity to the target. In the amplification step, the remaining target-bound sequences are recovered and amplified using the polymerase chain reaction (PCR) method, and a new pool of aptamers is created from the amplified PCR product. Using the new aptamer pool, the above three steps are repeated for 5 to 15 rounds to enhance the affinity for the target. After several rounds of the SELEX process, the sequence and secondary structure of the resulting product are analyzed, and the affinity between individual sequences for the target is measured to obtain the specific aptamer. However, the classical SELEX process involves a large number of rounds, has a very long selection period, and has a low success rate. To improve these issues, various modified SELEX processes have been reported [17,18]. These diverse SELEX processes not only improve the aptamer selection process but also allow for the high flexibility, specificity, and affinity of aptamers, making aptamers an excellent molecular recognition element.

### 1.2. Aptamers in Biosensors

A biosensor is an analysis system consisting of a biological recognition element that can recognize a target biomaterial, a physicochemical signal converter that converts the interaction between the target material and the recognition element into a measurable signal, and a signal analysis system that allows the user to easily analyze the signal [19]. Aptamers are biological recognition elements with a high sensitivity and selectivity and are suitable for use in ultra-sensitive biosensors for biomedical applications. They have already been reported to detect the specific interaction of aptamers with various biological targets such as small molecules, proteins, tumor cells, viruses, and pathogens using various analysis methods such as fluorometry, colorimetry, electrochemistry, and surface-enhanced Raman spectroscopy (SERS) [20,21,22,23,24]. Table 1 summarizes the targets, sequences, and affinity for the targets of all aptamers and the detection limits of the aptasensors introduced in this review. Among these, aptasensors using fluorescence detection methods are the most commonly used aptasensor type because they are easy to design, have a short detection time, have a high sensitivity and selectivity, are cost-effective, and, especially, facilitate the real-time detection of in vivo targets [25,26]. However, since aptamers consist of nucleotides that do not have fluorescence properties, the addition of an external fluorescent probe is necessary to be able to exhibit changes in fluorescence properties when aptamers interact with targets. These aptasensors are mainly based on Förster resonance energy transfer (FRET), fluorophore-linked aptamer assays (FLAA), fluorescent light-up aptamers (FLAPs), and fluorescence anisotropy (FA). Fluorescence-based aptasensors consist of an aptamer that recognizes a target, a fluorescent probe that generates a fluorescent signal, and sometimes nanomaterials added for signal amplification strategies.

This review will specifically focus on the developments and advances in the operation mechanisms of fluorescence-based aptasensors, the fluorescent probes, and fluorescent signal amplification strategies for biological applications. A schematic view of the scope to be summarized in this review is shown in Figure 1b.

## 2. Signal Generation Mechanisms

Since most targets are non-fluorescent, various fluorescence signal generation strategies, which will be described later, are used to detect them through fluorescence spectroscopy.

### 2.1. Förster Resonance Energy Transfer (FRET)

FRET is a process in which energy is transferred between an adjacent donor and acceptor fluorophore without the emission of light, resulting in the excitation of the acceptor fluorophore [27]. The FRET occurs when the transition dipole moment direction of this donor and receptor pair is correct and the emission spectrum of the donor and the absorption spectrum of the receptor overlap. The FRET process is exponentially affected by the change in distance (up to 10 nm) between two fluorophores. Due to this characteristic of FRET, which is very sensitive to distance, it has been widely applied in biosensors [28,29]. Since the structure of the aptamer indicates distinct changes before and after interaction with the target, the aptasensor modified with donor and acceptor fluorophores can quantitatively detect the target through fluorescent intensity changes (Figure 2a). For example, Lai et al. reported a study on detecting HepG2 cells, a human liver cancer cell line, through the TLS11a aptamer modified with the fluorophore FAM and the quencher Eclipse, which were designed to be normally folded into a hairpin structure [30]. The TLS11a aptamer was selected by Cell-SELEX through a mouse liver cancer cell line and is known to target liver cancer cells. Since this aptasensor is normally folded in a hairpin structure, the FAM and Eclipse are spatially close, so the FAM is quenched by the FRET process. After the aptamer binds to its target on the HepG2 cell surface, the distance between the FAM and Eclipse increases. As a result, the fluorescence signal of the FAM is recovered and HepG2 cells are detected. In this case, a design in which a fluorophore and a quencher are modified in one aptamer can detect the presence and location of the target, but the desired aptamer structural change before and after target binding must be strictly selected.

Suo et al. reported an aptasensor with improved design flexibility by modifying the phosphor and quencher into different ssDNA strands [31]. This aptasensor is a dual crossover DNA nanostructure modified with Cy3 and Cy5 and can simultaneously detect ochratoxin A (OTA) and aflatoxin B1 (AFB1), which are mycotoxins produced by fungi. Cy5 and Cy3 are modified with AFB1 and OTA aptamers, respectively, and these fluorophores are normally quenched by BHQ2-modified ssDNA that is complementary to the aptamer. In the presence of AFB1 or OTA, the aptamer for the target is deviated from the BHQ2-modified ssDNA and the fluorescence signal of Cy3 or Cy5 is recovered, enabling the simultaneous detection of both targets.

On the other hand, Xu et al. reported a FRET-based aptasensor for dopamine detection that does not depend on structural changes in the aptamer, using gold nanoparticles (AuNPs) as a quencher and free rhodamine B (RB) as a fluorophore [32]. In this aptasensor, the aptamer is employed as a ligand to improve the stability of AuNPs in the absence of a target, and non-aggregated AuNPs quench the fluorescence of RB. The aptamer bound to the target cannot act as a ligand; the addition of NaCl induces the aggregation of AuNPs and reduces the FRET efficiency between the aggregated AuNP and RB. This system has a simple design and allows for the quantitative analysis of targets without an additional modification of the aptamer, but the analysis value may be affected by the reaction environment and the ratio of AuNPs and RB. Li et al. reported a FRET-based aptasensor for detecting thrombin using [Ru(bpy)2(o-mopip)]^2+^ (OMO) as a fluorophore and graphene oxide (GO) as a quencher, without additional labeling of the aptamer [33]. In the absence of thrombin, the thrombin aptamer and OMO are adsorbed on GO by electrostatic and π-π interactions, and as a result, OMO is quenched. In the presence of thrombin, the aptamer binds to thrombin and forms a G-quadruplex. OMO binds to the newly formed G-quadruplex and dissociates from GO, and the fluorescence signal is recovered.

Quantification analysis using FRET between a fluorophore and a quencher relies on the signal intensity of a single fluorophore, whereas using FRET between a pair of fluorophores measures the fluorescence ratio between the donor and acceptor fluorophores, which is more advantageous for the background signal and absolute quantification. Sapkota et al. reported an aptasensor that detects lysozyme, a disease biomarker, through FRET between the donor Cy3 and the acceptor Cy5 [34]. Ben Aissa et al. reported an aptasensor that detects the quinolone antibiotic ofloxacin (OFL) through FRET between the donor FAM and the acceptor TAMRA [35].

### 2.2. Fluorophore-Linked Aptamer Assay (FLAA)

A Fluorophore-Linked Aptamer Assay is similar to ELISA, a conventional antibody-based assay also known as a sandwich-based assay, but differs in that it uses a fluorophore-linked aptamer instead of an enzyme-linked antibody [36]. This sandwich-type aptasensor consists of a primary aptamer that is immobilized on a platform and captures the target and a secondary aptamer that generates a signal when the target is present, and this pair of aptamers recognize the same target (Figure 2b). This sandwich-type detection strategy exhibits high sensitivity and specificity and is suitable for applying the lateral flow strip type kit for on-site diagnosis [37]. Franco-Urquijo et al. reported an aptasensor that can diagnose SARS-CoV-2 through the FLAA test [38]. They designed a sandwich-type detection system using C7/C9 aptamers that can specifically bind to the SARS-CoV-2 spike (S) protein. First, SARS-CoV-2 was captured by the C7 aptamer that was immobilized on a 96-well plate as a capture agent. After that, an FAM-labeled C9 aptamer was added, and the fluorescence signal was read from the well plate to detect SARS-CoV-2. This FLAA-based analysis platform can also be applied to small molecules such as antibiotics. Zhu et al. reported a study on detecting the antibiotic streptomycin (STR) in a glass flow cell using an FLAA-based aptasensor [39].

Applying this aptamer sandwich strategy with nanoparticles, it can be used for analysis to capture targets on the nanoparticle surface and separate them from the complex mixture. The advantage of this strategy is that it not only enriches the target material in the local area but also makes the target easy to purify. Xu et al. reported a study in which thrombin was separated from the mixture using silica nanoparticles (SNPs) modified with a primary aptamer and treated with a secondary aptamer labeled with carbon nanodots (C-Dots) to detect thrombin [40]. This aptasensor first binds thrombin to the surface of the aptamer-modified SNP to form an SNP–thrombin complex. The C-Dot-aptamer added later binds with thrombin on the surface of the SNP to form a fluorescent sandwich nanoparticle (SNP–Thrombin–C-Dot) and is separated from the unbound C-Dot-aptamer through centrifugation. The separated fluorescent sandwich particles were redispersed, and the detection signal was sensitively measured and quantified.

Using magnetic nanoparticles (MNPs), target substances can be separated more simply through magnetic separation instead of centrifugation. Bruno et al. reported a study on detecting various types of foodborne pathogens using surface-modified magnetic nanoparticles as a primary aptamer and a secondary aptamer labeled with quantum dots (QDs) [41]. In this aptasensor, the target bacteria and biotinylated aptamer were first mixed, and then this mixture was added to MNPs modified with streptavidin (SA). At this time, the MNP-target bacteria complex is formed through biotin–SA interaction. Afterwards, the MNP complex was separated from the mixture through magnetic separation, a QD-labeled aptamer is added, and fluorescence analysis is performed.

### 2.3. Fluorescent Light-Up Aptamers (FLAPs)

In 1999, an RNA aptamer that improved the fluorescence properties of malachite green (MG), a non-fluorescent organic molecule, was first discovered [42]. Although malachite green is cytotoxic and difficult in bioapplication, an aptamer that induces the fluorescence of a non-cytotoxic and non-fluorescent HBI derivative or thiazole orange derivative has been developed, enabling bioimaging and monitoring using FLAPs [43,44]. The advantage of FLAPs that exhibit fluorescence by combining with non-fluorescent molecules without external fluorescence probes is that they have low background signals. The principle of FLAPs is that the planar structure and conjugated system of the non-fluorescent small molecule bound to the aptamer are stabilized so that energy emission in the form of fluorescence emission becomes dominant over energy loss through non-radiative decay pathways such as molecular vibration or heat [45]. The detection of targets using FLAPs requires an allosteric type of aptamer that can induce the fluorescence of non-fluorescent small molecules and recognize the target (Figure 2c) [46,47].

Mou et al. reported an aptasensor that can detect cancer cells as well as analyze Cu^2+^ ions, an important trace element in living cells, by combining the AS1411 aptamer, which can specifically recognize cancer cells, and the Lettuce aptamer, an FLAP that induces the fluorescence of DFHBI-1T [48]. The AS1411 aptamer has a G-quadruplex structure with a G-rich sequence. In general, negatively charged nucleotides have low cell penetration efficiency, but the AS1411 aptamer targets nucleolin overexpressed in various cancer cells and is easily internalized into cancer cells. After that, the Letuce aptamer stabilizes the planar structure of the imidazolone and the phenyl rings of the bound DFHBI-1T to generate green fluorescent signals and allow for the detection of cancer cells. Cu^2+^ ions in the cell interact with the nucleobase of the aptamer to affect the secondary structure of the Lettuce aptamer, reducing the fluorescence signal of FLAPs and allowing for the real-time monitoring of intracellular Cu^2+^ ion abundance.

Endoh et al. reported an orthogonal FLAP aptasensor capable of simultaneously detecting multiple small molecules within living cells using different wavelengths of fluorescence [49]. The Orthogonal Light-Up Aptamer selected from the Broccoli aptamer derivative RNA library bound orthogonally to Isa-5a, DFHBI-1T, and malachite green, emitting blue, green, and red fluorescence, respectively. They showed that theophylline and S-adenosyl methionine (SAM) can be detected simultaneously through blue and green fluorescence in Flp-In 293 cells, and red fluorescence was measured as an indicator of transfection-positive cells capable of quantitative analysis. Sett et al. reported an FLAP aptasensor that detects theophylline by combining a light-up aptamer that generates a signal and a target binding aptamer through loop–loop interaction, instead of using an allosteric-type aptamer [50]. These two aptamers form a loop–loop connection only when the target is present, and when the target is not present, the binding of MG and a light-up aptamer is not allowed. This dual aptamer sensor linked by target-dependent loop–loop connection could be an alternative strategy for allosteric-type aptamers and easily expands the target types.

### 2.4. Fluorescence Polarization/Fluorescence Anisotropy (FP/FA)

FP/FA is one of the simple and powerful fluorescence signal generation approaches that can sensitively detect the interaction between aptamers and targets [51,52]. FP/FA can be obtained by measuring the fluorescence from a fluorescent probe excited with polarized incident light in the horizontal and vertical polarization planes and comparing the fluorescence intensities in the two polarization planes. The FP/FA value depends on the rotation speed of the fluorescent probe. If the probe is rotated at a high speed, the emitted fluorescence deviates from the polarization plane, resulting in a low FP/FA value. Conversely, if the probe rotation speed is slow, the fluorescence is still polarized and has a high FP/FA value. Since the rotation speed of the fluorescent probe is generally inversely proportional to the volume, the FP/FA value varies due to the volume change of the aptamer before and after the target binding [53].

Using this phenomenon, amplified FP/FA changes can be induced by increasing the volume change of the fluorescent probe before and after target binding (Figure 2d). For example, Li et al. reported improved FP/FA changes before and after target binding through a strategy of modifying the streptavidin protein with the DNA strand that hybridizes with an FAM-labeled aptamer [54]. In addition to this strategy of increasing the volume, an aptasensor that detects AFB1 by inducing a high FP/FA value employing a DNA sequence that interacts with dye was also reported [55]. This aptasensor used TAMRA as a fluorescent probe, and the DNA strand hybridizing with the TAMRA-labeled aptamer extends to the tandem G bases. After hybridization, the TAMRA is located spatially close to repeated G bases, and a high FP/FA value is induced by the interaction of the TAMRA and G bases. In the presence of AFB1, the aptamer is dehybridized from the G base-extended DNA strand by binding to AFB1, eliminating the TAMRA–G interactions, and resulting in a decrease in FP/FA values. Through these strategies, the aptasensor achieved the sensitive detection of small molecules such as AFB1, OTA, and adenosine triphosphate (ATP) using FP/FA analysis.

However, the sensitive detection of the target is possible without a signal amplifier in cases of detecting a large size of a target. In this case, a minimized aptamer is used to increase the detection sensitivity by increasing the difference in FP/FA values before and after target binding. Minagawa et al. reported an aptasensor that can detect human salivary *α*-amylase (sAA) and secretory immunoglobulin A (SIgA), markers of acute stress, from saliva samples through FP/FA analysis [56]. This aptasensor used a base-modified aptamer sequence to increase its affinity with the target. As another example of increasing detection sensitivity through FA/FP analysis, Zhao et al. reported an FP/FA-based aptasensor capable of detecting immunoglobulin E (IgE), an immunoantibody also used as a biomarker for various diseases [57]. In this aptasensor, the FAM was labeled at various positions of the aptamer to increase the difference in the FA/FP values before and after target binding.

## 3. Fluorescent Probes

As mentioned above, since aptamers do not have fluorescence properties, a fluorescence-based aptasensor system requires an external fluorescent probe. Fluorescent probes for fluorescence-based aptasensors for bioapplications include organic dyes and nanomaterials such as quantum dots and upconversion nanoparticles.

### 3.1. Organic Molecules as Fluorescent Probes

Fluorescent organic molecules (organic dyes) have the following advantages: (1) they can be easily modified into an aptamer, (2) they are inexpensive, and (3) a variety of natural and synthetic fluorophores are well known and can be selected depending on the purpose [58,59,60]. Previously developed organic dyes, fluorescein and rhodamine, are still used as fluorophores in aptasensors [30,32,35,38,54,57]. However, these previous fluorescent organic dyes have problems such as hydrophobicity, photobleaching, and being affected by pH changes [61]. Rhodamine derivatives such as TAMRA, TRITC, Texas Red, and cyanine dye families such as Cy2, Cy3, Cy5, and Cy7, which improve these problems and have brighter and more stable fluorescence properties compared to previous dyes, are widely used in aptasensors [31,33,35,55,62]. Additionally, Alexa and ATTO, which have improved fluorescence properties, have recently been developed to solve problems such as hydrophobicity, pH sensitivity, and photobleaching and are being used as fluorescent probes for aptasensors. For example, Setlem et al. and Cervantes-Salguero et al. reported an aptasensor using Alexa and ATTO as fluorophores, respectively [63,64].

Among these, fluorescent organic probes that emit near-infrared (NIR) light with a wavelength of 650–1000 nm are known to be particularly advantageous for biological applications [65]. NIR light is less absorbed by water, hemoglobin, melanin, and fat, which are the main absorbers of light in the body, so it has the maximum penetration depth into biological tissue, and the energy of light needed to excite the probe is low, so there is less risk of photodamage and photobleaching [66]. Shi et al. reported an aptasensor that is specifically recognized and accumulated in cancer cells and activates a fluorescent signal in an acidic extracellular environment, employing the NIR fluorescent dye Cy5 as a fluorescent probe (Figure 3a) [62]. This aptasensor is functionalized at the 5′ end of the aptamer that targets cancer cells, with a Cy5 dye–acid-labile acetal linker–BHQ2 quencher, designed to be activated in acidic environments. In a neutral pH environment, the Cy5 dye is located close to the BHQ2 quencher and is quenched, but in an acidic environment, the acetal linker is cleaved and the BHQ2 quencher is released from the Cy5 dye, restoring the fluorescence of Cy5. Since the extracellular acidic environment due to increased action and insufficient blood flow is one of the characteristics of cancer cells, the combination of linkers cleaving in response to the acidic environment and the aptamer targeting cancer cells enabled the imaging of cancer cells.

Various quencher organic molecules that can be used as FRET pairs with fluorescent organic molecules are also known [67,68]. The efficiency of the energy transfer process in FRET is affected not only by the distance between the donor fluorophore and the acceptor quencher but also by the degree of overlap between the emission spectrum of the fluorophore and the absorption spectrum of the quencher. In the design of fluorescence-based aptasensors utilizing FRET, it is important to select an appropriate donor fluorophore–acceptor quencher FRET pair with sufficient spectral overlap.

FLAPs, which emit fluorescence by combining with non-fluorescent organic molecules, can also be used as fluorescent probes. The main development directions of FLAPs for bioaplications are improved stability in the biological environment, the use of non-toxic, cell-permeable, non-fluorescent organic molecules, and the improvement and expansion of the fluorescence spectrum. 3,5-difluoro-4-hydroxybenzylidene imidazolinone (DFHBI) is one of the widely used non-fluorescent organic molecules in living intracellular imaging that meet these conditions. DFHBI is a mimic of the cyclic tripeptide, Ser65-Tyr66-Gly67, which is the chromophore of GFP and has a fluorescence peak at 501 nm when combined with Spinach or Broccoli aptamers. Similarly, the Pepper aptamer and Mango aptamer have affinity with TO1-Biotin (a derivative of Thiazole orange) and HPC620 (chemical analogs of Hexabenzocorone), respectively, which are non-fluorescent, non-toxic, cell membrane-permeable organic molecules, and can induce the fluorescence properties of these molecules. An intercalator dye, such as thiazole orange (TO) and SYBR green, that is intercalated into an oligonucleotide to improve fluorescence properties can also be used as a fluorescence probe for an aptasensor [69,70]. Mi et al. reported a bioluminescence resonance energy transfer (BRET)-based aptasensor that can detect antibiotics (tetracycline), metabolites (SAM), and signaling molecules (guanosine tetraphosphate) in live cells employing FLAPs, such as the Broccoli aptamer, Pepper aptamer, and Mango aptamer, as fluorescent probes (Figure 3b) [71]. BRET is similar to FRET, but it is different in using luciferase instead of fluorophore, which requires excitation light as a donor. Therefore, the BRET process does not require an external light source, and the background signal is very low and does not have issues such as autofluorescence and photo bleaching [72]. However, it has disadvantages such as a relatively weak signal intensity and short signal duration.

**Figure 3 molecules-28-07327-f003:**
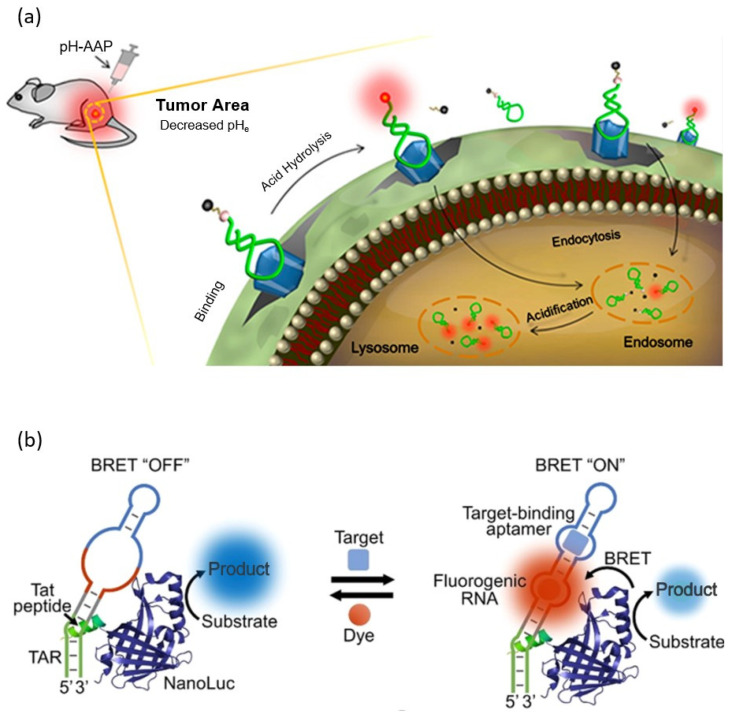
Schematic illustration of a fluorescence-based aptarsensor employing a (**a**) fluorescent organic molecule and a (**b**) non-fluorescent organic molecule. (Adapted from [62,71]).

### 3.2. Nanomaterials as Fluorescent Probes

Nanomaterials generally refer to materials in the form of particles with a diameter of 1–100 nm. Because nanomaterials have a very large surface area, they exhibit chemical, physical, electronic, and optical properties that are completely different from those of bulk materials [73]. Fluorescent nanomaterials such as quantum dots, upconversion nanoparticles, and fluorescent silica nanoparticles are attracting attention as an alternative for overcoming the disadvantages of organic molecules such as pH sensitivity and photobleaching properties.

Semiconductor quantum dots (QDs) are nano-semiconductor particles with a diameter of less than 10 nm. Semiconductor crystals reduced to a nano size have discontinuous quantized energy levels due to the quantum confinement effect [74,75]. Since the energy band gap between the conduction band edge and the valence band edge depends on the size of the QD, resulting in different emitted fluorescence wavelengths. QDs have several advantages, as follows. QDs consist of inorganic materials, have a high light stability, can have a variety of emission spectra even with the same composition depending on their size, have a wide absorption range, and are emitted with a narrow full-width-at-half-maximum (FWHM) and high color purity. QDs consist of a core/shell structure. Fluorescence emission occurs at the core, and the shell has a larger band gap than the core, so it prevents the loss of excited electrons and increases the quantum yield. Previously, due to the lack of technological alternatives and well-established synthetic processes, toxic cadmium-based QDs such as CdSe QD and CdTe QD were made biocompatible by covering them with an inorganic shell or capping them with organic ligands, and they have been widely used in the bioimaging and biosensing fields [76]. For example, Lian et al. reported an aptamer probe for detecting and imaging breast cancer cells such as MCF-7 and MDA MB 231, employing CdSe/ZnS QDs encapsulated with polyethylene glycol (PEG)-phospholipids functionalized with the AS1411 aptamer [77]. PEGylated lipids are mainly used to encapsulate cargo to increase the circulation time or prevent nonspecific adsorption, and in this aptasensor, they are used to stabilize CdSe/ZnS QDs and increase biocompatibility.

Recently, research has been conducted on the development of Cadmium-free QDs such as InP QD, ZnS-AgInS_2_ (ZAIS) QD, and Ag_2_S QD, which are less cytotoxic. Jin et al. reported an aptasensor that can detect the carbohydrate antigen CA125, known as a biomarker for various cancers, from real human body fluids using Ag_2_S QDs that emit NIR fluorescence as a fluorescent probe (Figure 4a) [78]. In this aptasensor, Ag_2_S QD is quenched by 5-fluorouracil (5-Fu) electrostatically adsorbed on the aptamer of CA125. In the presence of CA125, Ag_2_S QDs are released and the NIR fluorescence emission is restored because the aptamer/5-Fu complex has a higher affinity for CA125. Ayed et al. reported an aptasensor targeting the pathogen Acinetobacter baumannii (*A. baumannii*) using less toxic InP/ZnS QDs as fluorescent probes [79]. The aptamer with a high affinity to *A. baumannii* is conjugated to the surface of InP/ZnS QDs through EDC/NHS coupling, and this QD–aptamer probe specifically labels *A. baumannii*. Delices et al. reported non-toxic ZAIS QDs synthesized aqueously that can functionalize the surface with ssDNA [80]. This ZAIS QD emits 900 nm of NIR light and is expected to be a fluorescent probe that can be applied to living organisms, including for cell imaging.

Upconversion nanoparticles (UCNPs), another example of fluorescent nanomaterial that uses NIR lights as excitation light, have a high potential for applications in biomedical fields such as in vivo imaging and disease diagnosis due to their advantages such as a high light stability through the core/shell structure and a controllable emission wavelength [81,82]. UCNPs are nano-sized particles that exhibit the upconversion phenomenon of sequentially absorbing two photons of small energy and emitting light of larger energy. The upconversion phenomenon can occur through several mechanisms. Among them, the generally known Upconversion emission by the two photon absorption (TPA) process is inefficient and requires a high-powered light source because there is no energy level at which electrons can be excited between the ground state and the excited state. In contrast, the energy transfer upconversion (ETU) process allows electrons to be effectively excited step by step to a higher energy level by doping sensitizers that absorb long wavelengths of light and activators that have an excited meta-stable level between the ground and the excited state [83]. In general, Yb^3+^ and Nd^3+^ ions are doped as sensitizers, and Er^3+^, Tm^3+^, and Ho^3+^ ions are doped as activators. In addition to this advantage, research on UCNPs doped with lanthanide is actively being conducted due to their low bio-toxicity and their advantage of being excited in NIR light [84].

Liu et al. reported a fluorescent aptasensor that allows for the detection of Listeria monocytogenes (*L. monocytogenes*), an opportunistic foodborne pathogen, using aptamer-functionalized lanthanide-doped UCNPs and MNPs [85]. In this study, aptamer functionalized MNPs played a role in capturing and concentrating targets and enabled more sensitive detection. MNPs are an example of a nanomaterial that is highly biocompatible and is suitable for the magnetic separation and concentration of biological targets because most biological materials do not have magnetism, and they are widely applied in the biosensing field. Ouyang et al. reported another study on detecting Staphylococcus aureus (*S. aureus*), another foodborne pathogen, through an FRET-based aptasensor using UCNPs as fluorescent probes and AuNPs as quenchers (Figure 4b) [86]. AuNPs are functionalized with aptamers of *S. aureus*, and UCNPs are functionalized with cDNA (ssDNA, complementary with the aptamer). First, the AuNPs-aptamer is pre-hybridized with UCNPs-cDNA to quench the fluorescence of UCNPs. After that, the fluorescence of the UCNPs is restored and released due to the preferential binding of the aptamer and the added *S. aureus*. AuNPs are known to quench fluorescence very effectively through the process of surface energy transfer. In addition, because of its high biocompatibility and easy surface modification, it is a nanomaterial widely used in biological applications as a quencher and delivery agent [87]. In addition to AuNPs, graphene and MOF are nanomaterials widely used as quenching agents in FRET-based aptasensors [88,89].

Fluorescent silica nanoparticles (FSNPs) doped with organic dyes are another example of a fluorescent nanomaterial widely used in bioimaging and biosensors because they can reduce the problem of the photobleaching of organic molecules and are less cytotoxic [90]. FSNPs can be synthesized mainly by modified Stöber methods or reverse microemulsion methods, and the emission wavelength can be adjusted according to the type and ratio of the organic dye being doped [91]. Hu et al. reported a study on detecting HepG2 cells using a streptavidin-conjugated FSNP (SA-FSNP) and biotin-labeled TLS11a aptamer (Bio-TLS11a) (Figure 4c) [92]. First, Bio-TLS11a recognizes and binds with HepG2 cells and then treats SA-FSNPs to label FSNPs on HepG2 cells through biotin–streptavidin interaction.

**Figure 4 molecules-28-07327-f004:**
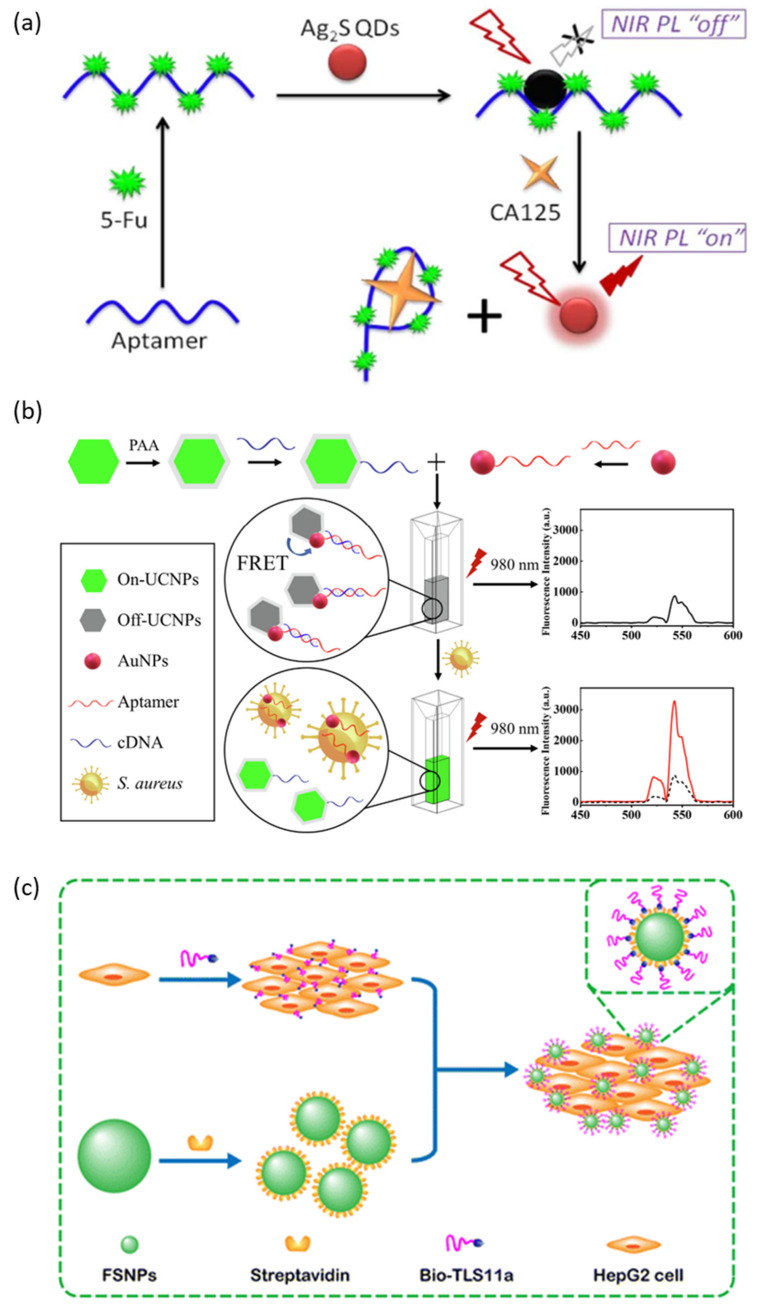
Schematic illustration of a fluorescence-based aptarsensor employing (**a**) a semiconductor quantum dot, (**b**) an upconversion nanoparticle, and (**c**) a fluorescent silica nanoparticle. (Adapted from [78,86,92]).

## 4. Signal Amplification Strategies

Fluorescence-based aptasensors can easily be conjugated with various fluorescence probes, targeting a wide range of materials with a high specificity and selectivity due to the flexibility of the aptamer, and also have the outstanding advantage of using various signal amplification strategies, including DNA-based amplification techniques such as CHA, HCR, RCA, PCR, and DNAzyme walker. In addition, since the aptamer is easily combined with signal amplification materials such as metal nanoparticles, mesoporous nanoparticles, and magnetic nanoparticles to amplify detection signals, many aptamer-based biosensors with a high sensitivity have been reported. Signal amplification strategies can be largely divided into three categories: DNA hybridization-based amplification strategies, enzymatic amplification strategies, and nanomaterial-based amplification strategies.

### 4.1. DNA Hybridization-Based Signal Amplification Strategies

Catalytic hairpin assembly (CHA) and hybridization chain reaction (HCR) are DNA hybridization-based amplification techniques that work through the toehold strand displacement of DNA strands. Since the rate constant of toehold strand displacement varies by a million times depending on the length of the toehold, the DNA hybridization-based amplification reaction is very fast and almost irreversible [93]. Because it works under isothermal conditions without external enzymes or thermal changes, it has a simple design and is easy to apply to a relatively diverse biological environment, and it has the advantages of having a high biocompatibility, amplification efficiency, being rapid and inexpensive.

CHA is an amplification technique that generates a large amount of double-stranded DNA by catalyzing the cross-opening of two complementary hairpin DNAs by an initiator [94]. Zhou et al. reported an aptasensor that detects cancer cell-derived exosomes with a high sensitivity by connecting a CHA amplification strategy and a protein-targeting aptamer (Figure 5a) [95]. The initiator is hybridized to an aptamer specific to the membrane protein of the exosome and immobilized on the surface of the magnetic bead. When the aptamer recognizes the membrane protein of the exosome, the aptamer captures the exosome with a higher affinity, releasing the initiator into the supernatant. The released initiator magnetic-separates from the magnetic bead and triggers the CHA circuit to amplify the fluorescent signal.

HCR is an amplification technique that generates very long double-stranded DNA by catalyzing the continuous self-assembly of two or more partially complementary hairpin DNAs by the initiator [96]. Yang et al. reported an aptasensor capable of sensitively and simultaneously detecting and imaging mucin 1 (MUC1) and nucleolin, biomarkers overexpressed in breast cancer cells, applying HCR amplification strategies (Figure 5b) [97]. Two independent HCR initiators were over-hanged at the vertices of the cell membrane-permeable and DNase-resistant tetrahedral DNA structure, each of which is hybridized with two recognition modules, the MUC1 aptamer and the AS1411 aptamer. When the aptamer is released by MUC1 or nucleoline, the activated initiator triggers an HCR circuit that amplifies different fluorescent probes (FAM, ZnPPIX/G4).

**Figure 5 molecules-28-07327-f005:**
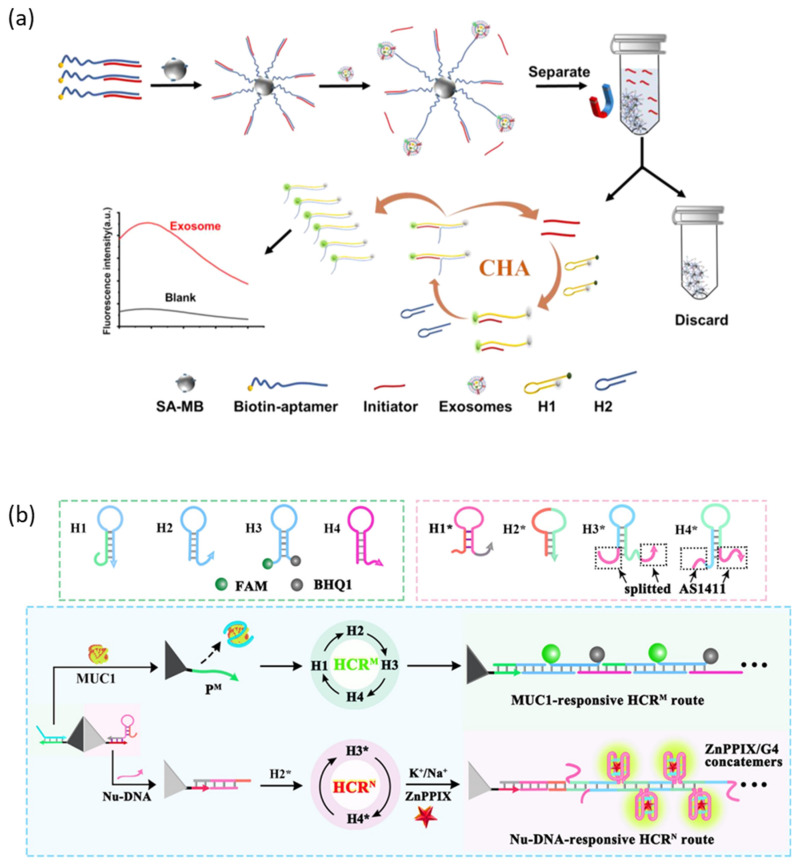
DNA hybridization-based signal amplification strategies. (**a**) Schematic illustration of the CHA amplification method. (**b**) Schematic illustration of the HCR amplification method. (Adapted from Refs. [95,97]).

### 4.2. Enzyme-Assisted Signal Amplification Strategies

Enzyme-assisted signal amplification strategies amplify signals using enzymes that catalyze biological reactions, and they can be largely divided into polymerase-assisted signal amplification strategies that amplify DNA strands and nuclease-assisted signal amplification strategies that cleave DNA strands. Among them, quantitative polymerase chain reaction (qPCR) is very sensitive and quantitative analysis technology that has been widely applied in the biomedical field since it was first developed [98]. PCR is a technology that replicates and amplifies the specific desired DNA template by repeating denaturation, annealing, and elongation cycles 25–50 times with Taq polymerase. In qPCR, the amount of DNA amplified at the end of each cycle is measured in real time, employing a fluorescent probe such as an intercalating dye or dye-labeled oligonucleotide. The amount of DNA that is amplified is almost doubled in each cycle, and the amount of the initial target template can be quantitatively analyzed through the measured amplification curve. Cavallo et al. reported an aptasensor that rapidly and accurately detects Leptin, a small protein that is secreted by adipose tissue and is closely related to obesity, by applying the qPCR signal amplification technique [99].

Rolling circle amplification (RCA) is an isothermal enzymatic DNA amplification technique in which phi29 DNA polymerase forms a long tandem repeat DNA structure from a circular DNA template [100]. Signal amplification through RCA is allowed by employing intercalating dyes or by the repeated formation of signal generation units such as DNAzyme. Bialy et al. reported an aptasensor capable of signal amplification through RCA that can detect platelet-derived growth factor (PDGF) and thrombin using a protein-targeting aptamer and intercalating dye that can act as primers (Figure 6a) [101]. In a normal state, aptamers are attached to a circular DNA template as primers, and RCA can occur when conditions for an RCA reaction are prepared. The amplified long product DNA strand by RCA combines with intercalating dye to produce a strong fluorescence signal. In contrast, the addition of a target protein induces the formation of a protein–aptamer complex, resulting in the aptamer not acting as a primer, inhibiting the RCA reaction and fluorescence signal generation. As another example, Wang et al. reported a sandwich-type aptasensor that amplifies the fluorescence signal through an RCA reaction to detect the SARS-CoV-2 antigen [102]. This aptasensor first captures a target using an antibody immobilized on a well plate and then forms an antibody-target–aptamer sandwich complex by the added aptamer. The aptamer is used as a primer for the subsequently added circular DNA template to enable the RCA reaction. The long product DNA strand formed by the RCA reaction amplified the detection signal of SYBR green, an intercalation dye.

Another isothermal enzymatic DNA amplification technology is Loop-mediated isothermal amplification (LAMP), which was developed relatively recently [103]. LAMP has the advantage of being simpler and easier to perform than PCR, a variable temperature amplification process, or RCA, which is relatively complex, and the amount of DNA amplified is larger than that for PCR. However, it has the disadvantage of limited application due to the difficulty of the primer design. The signal amplification process through LAMP uses four individual primer sets to recognize and amplify six individual sequence regions, resulting in high efficiency and selectivity, and the exponential amplification function improves detection sensitivity. Aubret et al. reported a sandwich-type aptasensor that amplifies the detection signal with LAMP, which consists of a magnetic bead modified with a primary aptamer and a secondary aptamer with an extended dumbbell sequence for the LAMP process [104].

The polymerase-assisted signal amplification methods can detect the target sensitively, but specific primers need to be designed, whereas the nuclease-assisted signal amplification methods have attracted great attention because they can detect a target with high sensitivity and easy operation [105]. Among them, exonuclease III (Exo III), a sequence-independent nuclease, is a widely used Nuclease in signal amplification strategies because it catalyzes the step-by-step hydrolysis of mononucleotides at the blunt or recessed 3’-end of double-stranded DNA without recognizing a specific sequence. For example, Liu et al. reported an aptasensor that sensitively detects OTA by applying an Exo III-assisted signal amplification strategy (Figure 6b) [106]. The aptamer of OTA is pre-hybridized with cDNA, which can trigger the Exo lll-assisted amplification circuit, and when OTA is recognized, it binds with OTA and releases cDNA. The released cDNA is hybridized with folded hairpin H, and then the blunt 3′ end of the double-stranded hairpin H/cDNA is hydrolyzed by Exo lll, and the cDNA is re-released. This circuit repeats and the deactivated part of the hairpin H is activated and amplified. The amplified short-stranded DNA captures a quenched fluorescent probe that is adsorbed onto graphene with π–π stacking interactions to recover the fluorescence signal. In addition, studies have been reported in which the sensitivity of the aptasensor was improved by amplifying the detection signal with a Duplex-specific nuclease (DSN) that specifically cleaves double-stranded DNA or DNase I, an endonuclease that non-specifically cleaves single-stranded and double-stranded DNA [105,107].

Clustered Regularly Interspaced Short Palindromic Repeats (CRISPR)/CRISPR-associated (Cas) nuclease is a gene editing tool that cleaves specified DNA sequences derived from the adaptive immune system of bacteria and archaea [108]. The CRISPR Cas system consists of CRISR single guide RNA (sgRNA) that specifies target DNA and Cas nuclease that cleaves DNA, and this controllable sequence-specific recognition ability has been proven to be a rapid diagnostic tool for detecting target DNA [109]. In addition to sequence-specific dsDNA cleaving, Cas12a has been reported to activate non-specific ssDNA cleaving upon complementary ssDNA binding to CRISPR RNA (crRNA) [110]. Employing these properties of Cas12a, Peng et al. reported an aptasensor combined with CRISPR-Cas12a for the sensitive detection of ATP (Figure 6c) [111]. In this strategy, the ATP aptamer prehybridizes crRNA as target DNA and induces the non-specific cleavage of reporter ssDNA double-labeled with BHQ1 and FAM in the absence of ATP. In the presence of ATP, the ATP–aptamer complex is formed, reducing the amount of target DNA that Cas12a can process and inhibiting the cleaving of reporter ssDNA. The intensity of the fluorescence signal generated by the cleaving of the reporter ssDNA has a linear correlation with the concentration of ATP. Feng et al. reported a sandwich-type aptasensor platform that detects cardiac troponin I (cTnI), a biomarker of myocardial damage, as a model target by applying a signal amplification strategy that effectively combines CRISPR Cas 13d and T7 RNA polymerase (T7 RNAP) [112]. CRISPR Cas13d is a recently identified CRISPR Cas system that knocks down RNA by targeting and cleaving RNA. This aptasensor forms a sandwich complex only when the target is present, allowing subsequent T7 RNAP to transcribe the trigger RNA. As a result, Cas13d is activated by the transcribed trigger RNA and catalyzes the cleavage of the reporter RNA modified with a fluorophore and a quencher to amplify the detection signal.

DNAzyme walker is a signal amplification technology using another enzymatic oligonucleotide cleavage reaction that combines the catalytic cleavage ability of DNAzyme and DNA walker [113]. The general fluorescence signal amplification strategy using DNAzyme walker generates a fluorescence signal by autonomously cleaving and walking along a fluorescent labeled substrate DNA track located spatially close to the quencher. For example, Yang et al. reported an aptasensor capable of detecting the ultra-high sensitivity of kanamycin through a DNAzyme walker-based signal amplification strategy (Figure 6d) [114]. The Kanamycin aptamer is pre-hybridized with cDNA labeled with FAM, functionalized on the magnetic bead, to protect cDNA from DNAzyme walker. In the presence of kanamycin, the aptamer specifically binds with kanamycin and is released from cDNA. After that, DNAzyme walker is hybridized with cDNA, and DNAzyme is activated by the cofactor Pb^2+^ ion, cleaves cDNA, and releases an FAM-labeled short DNA fragment into the supernatant. The fluorescence signal of the FAM present in the supernatant allowed for the quantitative analysis of kanamycin by removing the FAM not released from the magnetic beads through magnetic separation.

**Figure 6 molecules-28-07327-f006:**
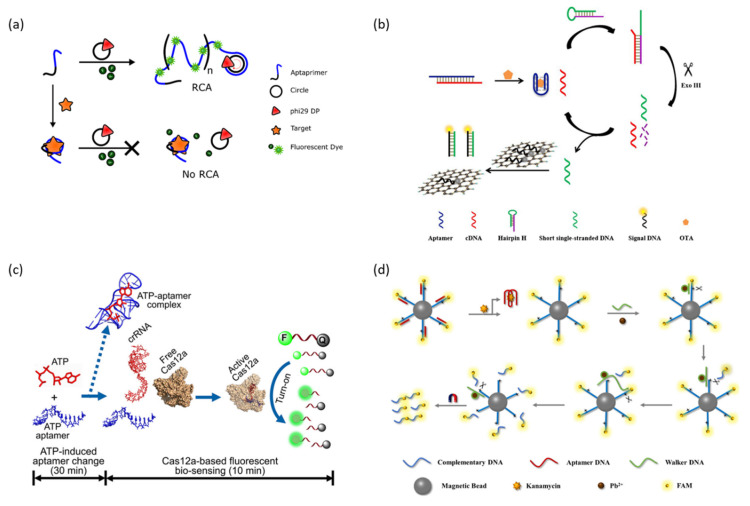
Enzyme-assisted signal amplification strategies. (**a**) Schematic illustration of the rolling circle amplification method. (**b**) Schematic illustration of the exonuclease III-assisted method. (**c**) Schematic illustration of the CRISPR Cas12a-assisted method. (**d**) Schematic illustration of the DNAzyme walker-assisted method. (Adapted from Refs. [101,106,111,114]).

### 4.3. Nanomaterial-Based Signal Amplification Strategies

Nanomaterial-based signal amplification strategies are (1) methods of concentrating a target material through magnetic separation employing MNPs, (2) methods of controlled fluorescent dye release employing dye-loaded porous nanomaterials, and (3) methods of a metal-enhanced fluorescent that enhances the fluorescence intensity of the fluorophore with the surface plasmon resonance employing metal nanomaterials.

As mentioned above, because most biological materials do not have magnetism, magnetic separation can easily separate and concentrate targets from complex biological samples. In addition, MNPs are widely used in various analytical applications due to their advantages such as their high stability, homogeneity, biocompatibility, rapid response to magnetic fields, and easy surface modification [115]. For example, Wang et al. reported an analysis platform that isolates exosomes that have CD63 marker proteins on the surface employing aptamer-functionalized MNPs that target CD63 proteins and is applicable to various downstream analyses through controlled elution (Figure 7a) [116]. In this analysis platform, the aptamer plays a role in capturing and immobilizing the CD63 protein onto MNP and to control elution after magnetic separation. Controlled elution was possible by changing the tertiary structure of the aptamer by adding NaCl salt, reducing the affinity between the aptamer and target, and resulting in the release of the captured target.

Mesoporous silica nanoparticles (MSNPs) have been widely applied in controlled release systems for drugs as well as fluorescent dyes due to their excellent properties such as their high thermal stability, biocompatibility, excellent tunable porosity, and large loading capacity [117]. The release of the cargo can be controlled by pH change, reduction and oxidation, light irradiation, etc. In particular, systems designed to release cargo by the recognition of the target can be applied to target drug delivery systems and signal amplification systems for target detecting. Tan et al. reported an aptasensor capable of detecting AFB1 using MSNP loaded with Rhodamine 6G (Rh6G) and an aptamer with high affinity for the target as gate molecules. (Figure 7b) [118]. In this aptasensor, the aptamer is adsorbed onto the MSNPs by electrostatic interaction and blocks the pore. Because the aptamer has a higher affinity for AFB1 than for the MSNP, the pore opens in the presence of AFB1 and releases the loaded Rh6G.

Metal-Enhanced Fluorescent (MEF) is a phenomenon in which the fluorescence of a fluorophore located spatially close to the metal nanoparticles such as gold (Au), silver (Ag), and platinum (Pt) is enhanced by the localized surface plasmon resonance (LSPR) energy of the metal nanoparticles [119]. When an electromagnetic wave (light) of a specific frequency irradiates to a metal nanoparticle, the conduction electrons of the nanoparticles collectively oscillate by interaction with the electromagnetic field of light, resulting in LSPR. Coupling with a strong electromagnetic field generated locally near the nanoparticle modified the absorption properties of the fluorophore, which resulted in an increased fluorescence intensity and quantum yield, decreased lifetime, and increased rate of excitation and emission, resulting in increased photostability [120]. Metal-enhanced fluorescence (MEF) is an efficient phenomenon that amplifies fluorescence signals, and various MEF-based biosensors have been developed and studied. Pang et al. reported an MEF-based fluorescent aptasensor by combining an aptamer targeting hemagglutinin (HA) protein, an antigenic spike of the avian influenza virus H5N1, with MEF employing Ag@SiO_2_ NPs (Figure 7c) [121]. In this aptasensor, the fluorescence signal is generated by the intercalating of the TO molecule into the aptamer functionalized on the Ag@SiO2 NPs and is amplified by the MEF phenomenon. In the absence of HA, the aptamer remains in the form of a single strand in which TO molecule intercalation does not occur, and the free TO molecule showed almost no fluorescence. In the case of HA present, the aptamer combines with HA to form an HA/aptamer G-quadruplex complex with which TO molecules can be intercalated. TO molecules intercalated with the G-quadruplex generate amplified fluorescence signals by the surface plasmon resonance enhancement of Ag@SiO_2_ NPs.

Table 2 summarizes the merits and demerits of each signal amplification strategy introduced in Section 4.

**Figure 7 molecules-28-07327-f007:**
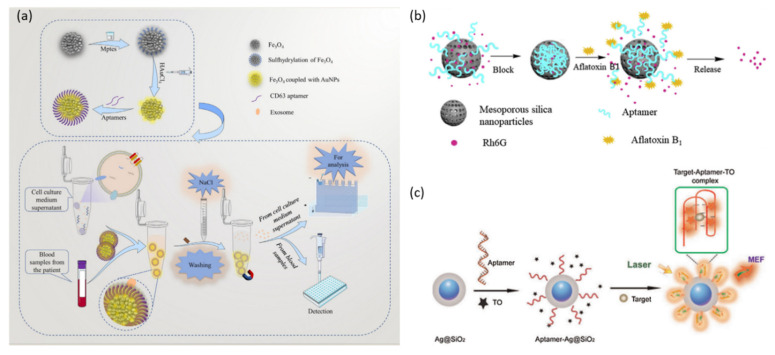
Nanomaterial-based signal amplification strategies. (**a**) Schematic illustration of target purification and concentration through magnetic separation. (**b**) Schematic illustration of signal amplification through the controlled release of dye employing dye-loaded mesoporous silica nanoparticles. (**c**) Schematic illustration of signal amplification through metal-enhanced fluorescence. (Adapted from Refs. [116,118,121]).

**Table 1 molecules-28-07327-t001:** Summary of all aptamers introduced in this review.

Target	Sequence Details (5′-3′)	Affinity	Limit of Detection	Ref.
Anatoxin-a	TGG CGA CAA GAA GAC GTA CAA ACA CGC ACC AGG CCG GAG TGG AGT ATT CTG AGG TCG G	K_d_: 81.3 nM	4.45 pM	[20]
PTK 7	ATC TAA CTG CTG CGC CGC CGG GAA AAT ACT GTA CGG TTA GA	K_d_: 0.78 nM	0.016 ng/mL	[21]
SARS-CoV-2	CGC AGC ACC CAA GAA CAA GGA CTG CTT AGG ATT GCG ATA GGT TCG GTT TTT	K_d_: 43 ± 4 nM	7 nM	[22]
	ATC CAG AGT GAC GCA GCA AGG GTA TTG GCA GTG GTA GGT ACT GCG TGC GTT GTG GTT CTA GCA TGT TTA ATG GAC ACG GTG GCT TAG T	K_d_: 21.2 nM	10^6^ copies/mL	[37]
	CAC GTG GCC CAC GTT AAT CCG TTA TAA GTC AAG CTC GAT	K_d_: 89.41 ± 18 nM	41.87 nM	[38]
	GGG GGC GTC AAG CGG GGT CAC ATC GGA GTA GGG AAT CTT G	K_d_: 231.9 ± 15 nM		
	TTC CGG TTA ATT TAT GCT CTA CCC GTC CAC CTA CCG GAA TTT TTT TTT TTT TTT TTT TTT TTT TTT TTT ACG GGT TTG GCG TCG GGC CTG GCG GGG GGA TAG TGC GGT	K_d_: 0.51 nM	5.1 × 10^4^ TU/mL	[102]
Hep G2 cell	ACA GCA TCC CCA TGT GAA CAA TCG CAT TGT GAT TGT TAC GGT TTC CGC CTC ATG GAC GTG CTG	K_d_: 4.51 ± 0.39 nM	~100 cells/ml	[30]
	AGT AAT GCC CGG TAG TTA TTC AAA GAT GAG TAG GAA AAG A	-	-	[90]
Ochratoxin A	GAT CGG GTG TGG GTG GCG TAA AGG GAG CAT CGG ACA	K_d_: 63 ± 18 nM	0.0058 ng/mL	[31]
			1 nM	[54]
				[106]
	GGC AGT GTG GGC GAA TCT ATG CGT ACC GTT CGA TAT CGT G	K_d_: 290 ± 150 nM	9 nM	[69]
	GGC GCA TGA TCA TTC GGT GGG TAA GGT GGT GGT AAC GTT G	K_d_: 110 ± 50 nM	-	
Aflatoxin B1	CAC GTG TTG TCT CTC TCT GTG TCT CGT G	K_d_: 27.7 ± 2.4 nM	0.046 ng/mL	[31]
	TGC ACG TGT TGT CTC TCT GTG TCT CGT GC	-	60 pM	[54]
	TTC TTC TGG CTT GGT GGT TGG TGT GTC TGC TGA TTT GGT A	K_d_: 50.45 ± 11.06 nM	20 ppb	[62]
	ATC CGT CAC ACC TGC TCT GAC GCT GGG GTC GAC CCG	Kd: 35.6 ± 2.9 nM	0.05 μg/kg	[88]
	AGT TGG GCA CGT GTT GTC TCT CTG TGT CTC GTG CCC TTC GCT AGG CCC ACA	-	0.13 ng/mL	[118]
Dopamine	GTC TCT GTG TGC GCC AGA GAA CAC TGG GGC AGA TAT GGG CCA GCA CAG AAT GAG GCC C	K_d_: 1.6 μM	2 nM	[32]
Thrombin	AGT CCG TGG TAG GGC AGG TTG GGG TGA CT	K_d_: 6 nM	0.76 nM	[33]
			1 nM	[40]
			100 pM	[104]
	GGT TGG TGT GGT TGG	K_d_: 5 nM	1 nM	[40]
			240 pM	[101]
			100 pM	[104]
Lysozyme	ATC AGG GCT AAA GAG TGC AGA GTT ACT TAG	K_d_: 65 nM	30 nM	[34]
Ofloxacin	AAG TGA GGT TCG TCC CTT TAA TAA ACT CGA TTA GGA TCT CGT GAG GTG TGC TCT ACA ATC GTA ATC AGT TAG	K_d_: 56.9 ± 11.3 nM	0.12 μM	[35]
Streptomycin	CGG CAC CAC GGU CGG AUC	-	33 nM	[39]
	GAU CGC AUU UGG ACU UCU GCC	K_d_: 1 μM		
ATP (w/MG)	UCC CGA CUG GGG GAG TAT TGC GGA GGA AGG UAA CGA AUG GA	K_d_: 50 μM	10 μM	[46]
TH (w/MG)	GUC GUA ACG AAU GGA UAC CAU GCA UGC ACC UUG GCA GCC CGA GAC	K_d_: 40 μM	2 μM	
FMN (w/MG)	GCG GUA ACG AAU GUA GGA UAU GCA UGA UGC AGA AGG ACC GAC GC	K_d_: 30 μM	-	
DFHBI-1T (for Cu^2+^ ion)	CTT AGT AGG GAT GAT GCG GCA GTG GGC TTC ATC TAT ATA AGA TGA GGG GAC TAA G	K_d_: 223.6 nM (w/o Cu^2 +^) 175.6 μM (w/Cu^2+^)	0.1 μM (Cu^2+^)	[48]
Nucleolin	GGT GGT GGT GGT TGT GGT GGT GGT GG	K_d_: 16.36 ± 10.30 nM	-	[48]
			-	[77]
			0.87 pM	[97]
Isa-5a (for Theo)	GGU ACC GGA AUC UGU CGA GUA GAG UGU GGU CGA UAC CAG CCG AAA GGC CCU UGG CAG CGA AGG UCG GGU CCA GAU ACC GGU GCC	K_obs_: 0.337 ± 0.0259 nM^−1^ (no Theo) 2.27 ± 0.953 nM^−1^ (20 μM Theo)	782 nM (Theo)	[49]
DFHBI-1T (for SAM)	GGU ACC GGA AUC UGU CGA GUU GGA GUG UGG UCC GAA AGG AUG GCG GAA ACG CCA GAU GCC UUG UAA CCG AAA GGG GAA GGU CGG UUC CAG AUA CCG GUG CC	K_obs_: 0.23 ± 0.04 μM^−1^ (no SAM) 2.25 ± 0.03 μM^−1^ (10 μM SAM)	301 nM (SAM)	
Theophylline	GGC GAU ACC AGC ACU GGG AAG CCC UUG GCA GCG UC	-	-	[50]
Adenosine	CCT GGG GGA GTA TTG CGG AGG AAG G	K_d_: 6 ± 3 μM	1 μM	[53]
ATP	ACC TGG GGG AGT ATT GCG GAG GAA GGT	K_d_: 31 ± 3 μM	0.5 μM	[54]
			400 nM	[111]
SIgA	AAT CTC CCT AAT CTG CTG ATG TTT GTA TTT CAA ATT	K_d_: 10.4 nM	-	[56]
Salivary α-amylase	ATT GTG AAC GAC GTG AAT AGT GTT TGT GGG TCC GGA GTT	K_d_: 441 pM	-	
Immunoglobulin E	GGG GCA CGT TTA TCC GTC CCT AGT GGC GTG CCC C	K_d_: 0.8 nM	20 pM	[57]
Cortisol	GCCCGCATGTTCCATGGATAGTCTTGACTA	-	-	[63]
A549 cell	GTG GCC AGT CAC TCA ATT GGG TGT AGG GGT GGG GAT TGT GGG TTG	K_d_: 94.6 nM	-	[66]
Tetracycline	CGT ACG GAA TTC GCT AGC CCC CCG GCA GGC CAC GGC TTG GGT TGG TCC CAC TGC GCG TGG ATC CGA GCT CCA CGT G	K_d_: 63.6 nM	0.029 μg/mL	[70]
	AAA ACA UAC CAG AUU UCG AUC UGG AGA GGU GAA GAA UAC GAC CAC CU	K_d_: 1 μM	0.1 μM	[71]
Tat peptide	GGC UCG UUG AGC UCA UUA GCU CCG AGC C	K_d_: ~10 nM	-	
S-Adenosylmethionine	GAA AGG AUG GCG GAA ACG CCA GAU GCC UUG UAA CCG AAA GG	K_d_: 1.7 μM	1 μM	
Guanosine tetraphosphate	CAG CGA CCG AGC GGU ACA A / ACA CCG UGA GCA UAA AAG GCU CCA	K_d_: 10 nM	1 μM	
CA125	AAA AAU GCA UGG AGC GAA GGU GUG GGG GAU ACC AAC CGC GCC GUG	K_d_: 4.13 nM	0.07 ng/mL	[78]
Acinetobacter baumannii	TAC ATG GTC AAC CAA ATT CTT GCA AAT TCT GCA TTC CTA CTG T	K_d_: 7.547 ± 1.353 pM	~0.5 × 10^8^ cells/mL	[79]
Listeria monocytogenes	GGG AGC TCA GAA TAA ACG CTC AAT ACT ATC GCG GGA CAG CGC GGG AGG CAC CGG GGA TTC GAC ATG AGG CCC GGA TC	K_d_: 48.74 ± 3.11 nM	8 CFU/mL	[85]
Staphylococcus aureus	GCA ATG GTA CGG TAC TTC CTC GGC ACG TTC TCA GTA GCG CTC GCT GGT CAT CCC ACA GCT ACG TCA AAA GTG CAC GCT ACT TTG CTA A	-	10.7 CFU/mL	[86]
Chloramphenicol	ACT TCA GTG AGT TGT CCC ACG GTC GGC GAG TCG GTG GTA GCC C	-	0.09 nmol/L	[89]
	CAC CCC ACC TCG CTC CCG TGA CAC TAA TGC TA	K_d_: 17.1 nM	-	[116]
CD63	ATA TAC ACC CCA CCT CGC TCC CGT GAC ACT AAT GCT A	-	0.5 particles/μL (EXO-MCF-7) 0.1 particles/μL (EXO-PANC-1)	[95]
EpCAM	CAC TAC AGA GGT TGC GTC TGT CCC ACG TTG TCA TGG G	K_d_: 22.8 ± 6.0 nM		
Mucin 1	GCA GTT GAT CCT TTG GAT ACC CTG G	K_d_: 38.3 nM	0.75 nM	[97]
Leptin	GTT AAT GGG GGA TCT CGC GGC CGT TCT TGT TGC TTA TAC A	K_d_: 1.5 ± 0.25 μM	100 pg/mL	[99]
PDGF	AAG GCT ACG GCA CGT AGA GCA TCA CCA TGA TCC TG	K_d_: ~0.1 nM	6.8 nM	[101]
Saxitoxin	CTT TTT ACA AAA TTC TCT TTT TAC CTA TAT TAT GAA CAG A	K_d_: 61.44 ± 23.18 nM	0.035 ng/mL	[105]
Cardiac troponin I	CGT GCA GTA CGC CAA CCT TTC TCA TGC GCT GCC CCT C	K_d_: 270 pM	12.6 pM	[112]
	CGC ATG CCA AAC GTT GCC TCA TAG TTC CCT CCC CGT GTC C	K_d_: 317 pM		
kanamycin	TGG GGG TTG AGG CTA AGC CGA	K_d_: ~78.8 nM	0.00039 ng/mL	[114]
H5N1	TTG GGG TTA TTT GGG AGG GCG GGG GTT	K_d_: 24.7 nM	2 ng/mL	[121]

**Table 2 molecules-28-07327-t002:** Comparison of the introduced signal amplification strategies.

Amplification Strategies	Merits	Demerits
CHA	Simple design and fewer non-specific reactions.	Less sensitivity due to the low reaction efficiency.
HCR	High sensitivity due to the high amplification efficiency.	High background due to the non-specifically triggerd opening of the probe.
RT PCR	Real-time detection during exponential amplification.	Requires expensive equipment for variable temperature nucleic acid amplification and real-time monitoring.
RCA	Isothermal nucleic acid amplification reaction that does not require separate equipment. High sensitivity due to the high amplification efficiency.	Largely affected by the purity of the circular template. Non-specific binding in complex environments due to the large size of the product.
LAMP		Various applications are limited due to difficulties in the primer design.
Nuclease assisted strategies	Specific amplified signal detection due to the specific site recognition ability and high catalytic efficiency of nuclease.	Nucleases are expensive, have low stability, and are difficult to preserve.
DNA walker	High directionality, flexibility, accuracy, sensitivity, and efficiency.	Limited applications due to complex systems consisting of DNA walker, DNA track, and driving force.

## 5. Conclusions

We summarized the broad applications of fluorescence-based aptasensors for disease diagnosis and prevention, as well as their main signal generation mechanisms and signal amplification strategies. The fluorescence-based aptasensor is capable of the rapid, sensitive, accurate, cost-effective, and non-invasive detection of various targets such as heavy metals, small molecules, biomolecules, pathogenic bacteria, biotoxin, antibiotic residues, and viruses, making it an excellent analytical tool widely used in applications for quantitative analysis, disease diagnosis, real-time dynamic monitoring of living cells, and food safety detection. Aptamer-based sensors are widely studied for the industrialization of food and the prevention of diseases through rapid food safety analysis as well as disease diagnosis and clinical application through various clinical-related biomarker analyses.

In addition, aptamers have a lower immunogenicity and wider target range than antibodies, so they were considered an excellent alternative to antibodies in clinical applications. However, despite the advantages of aptasensors, clinical and in vivo application is still very limited. This may be because most aptamers were selected in an in vitro environment and are thus greatly affected by the complex environment in vivo. In vivo, there are various DNA or RNA restriction enzymes that can degrade the structure of aptamers, the interaction between the target and aptamer can vary depending on the temperature, pH, ions, and biomolecules in the living body, and kidney filtration is fast due to the small size of aptamers [122]. This decline in the in vivo utilization of aptamers is one of the main factors interfering with in vivo application. Although many studies have been reported for solving the problems of aptamers and improving stability and utility in vivo, such as the use of modified nucleotides, functionalization with nanomaterials, and strategies for maintaining affinity for in vivo targets through in vivo SELEX, aptamers have not yet replaced antibodies in clinical settings.

From this perspective, aptamers may have tremendous advantages over antibodies in fluorescence bioanalysis applications. Fluorescence bioanalysis had problems such as high background signals caused by autofluorescence and a shallow penetration depth. Because aptamers have the excellent property of changing their tertiary structure when binding with a target, they could be combined with various signal generation and signal amplification strategies to solve traditional problems. Aptamers can be selected through the SELEX process and can apply various modified SELEX processes as required, allowing for more flexible design. The SELEX process, which selects aptamers with a high specificity and affinity, continues to develop, shortening the selection period and increasing the success rate, and can be differentiated depending on the purpose. In addition, aptamers have an important advantage over antibodies in that they can be easily functionalized with a variety of fluorescent probes, ranging from naturally derived organic fluorescent dyes to synthetic fluorescent nanoparticles. The optical and physical properties of these fluorescent probes have been well defined, and methods for modifying them with aptamers are well known.

The future development trends of aptasensors can be summarized from two perspectives. The first is the improvement and cascade combination of various signal amplification strategies. Signal generation and amplification strategies are currently being continuously improved, such as reducing background signals and false positives, improving linear responses to nonlinear responses, and exponentially amplifying detection signals. In addition, two or more signal amplification strategies can be effectively cascade-combined to significantly increase sensitivity. The second is the development of an analysis platform that can perform on-site analysis quickly and easily by combining it with a microfluidic analysis device. In particular, the food safety and biomedical fields require quick analysis systems capable of the rapid detection of pathogens or viruses.

Aptamers are developing rapidly, overcoming previous problems, and more stable, useful, and biocompatible fluorescent probes and improved signal generation and amplification strategies for sensitive and accurate detection are being reported. Therefore, due to the flexibility of aptamers applicable to various targets, fluorescent probes, signal generation, and amplification strategies, fluorescence-based aptasensors are expected to be a next-generation detection technology for promoting and developing research for intracellular monitoring, disease diagnosis, and prevention in various fields including biotechnology, biomedicine, and food safety.

## Figures and Tables

**Figure 1 molecules-28-07327-f001:**
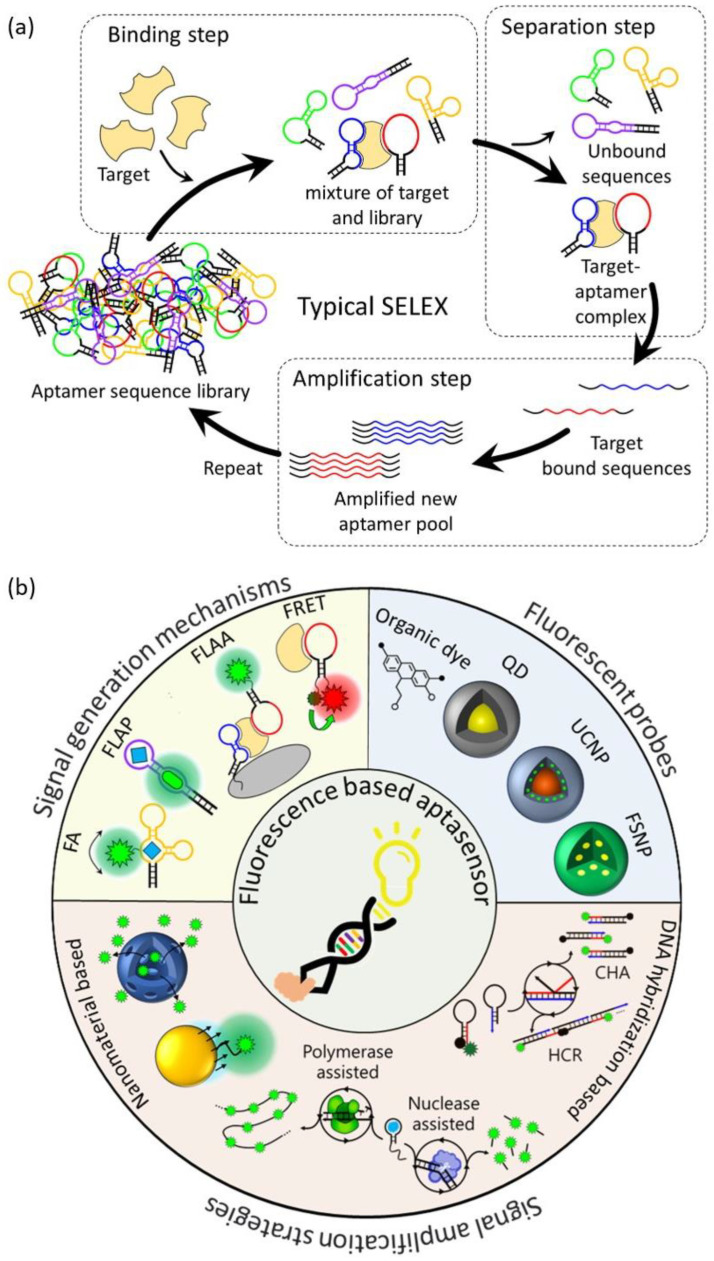
Schematic illustration of (**a**) a typical SELEX protocol for aptamer selection, (**b**) aptamer-based fluorescent biosensors for biological application.

**Figure 2 molecules-28-07327-f002:**
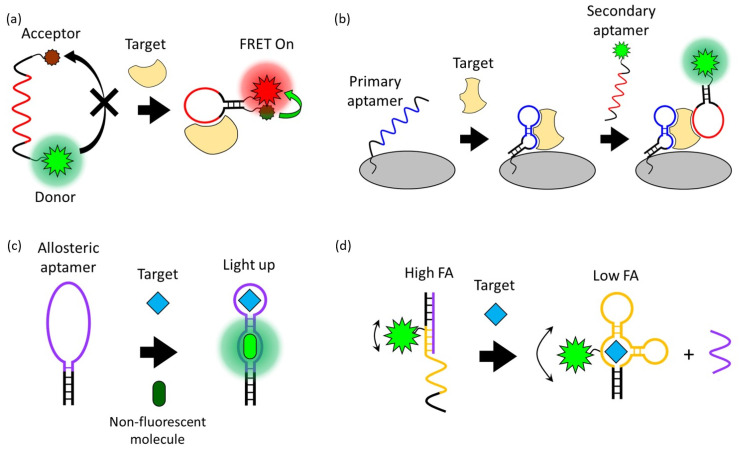
Schematic overview of signal generation mechanisms based on (**a**) Förster resonance energy transfer, (**b**) a fluorophore-linked aptamer assay, (**c**) fluorescent light-up aptamers, and (**d**) fluorescence polarization/fluorescence anisotropy.

## Data Availability

The data presented in this study are available upon request from the corresponding author. The data are not publicly available due to ethical considerations.

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
