# Peer review of "Recent Advances in Biological Applications of Aptamer-Based Fluorescent Biosensors"

_molecules, 2023, doi:10.3390/molecules28217327_

Round 1

Reviewer 1 Report

Comments and Suggestions for Authors

This review summerized the recent advances of fluoresence aptasensors in biological application.  They introduced the signal generation mechanisms, fluorescent probes, signal amplification strategies and detection molecules of fluorescent aptasensors. This topic is broad and interesting, which will attract reader's attention. It can be accepted after revision. 

1.  Figure 1b should be revised. I can't obtain any information of fluorescent aptasensor in biological application.  The detection molecules should be added in Figure 1b.

2. The  effect of experimental conditions (sucn as pH, temperature, concentration) on the analytical performance of fluorescent aptasensors should be discussed.

3.  The prospects and challenges of fluorescent aptasensors should be deeply discussed. 

4.  Many grammer mistakes should be checked. The format of references should also be checked.

Comments on the Quality of English Language

 Minor editing of English language required

Reviewer 2 Report

Comments and Suggestions for Authors

The review article "Recent advances in biological applications of aptamer-based fluorescent biosensors" addresses an interesting, complex, and highly topical subject. The article contains valuable and importat information. However, the subsections contain a lot of theoretical concepts and too few personal opinions. In this format, it seems closer to a chapter in a book rather than a review article. In my opinion, the main improvement would consist of providing clear and more numerous examples of the approaches reported in the literature at this time. I believe the article could be successfully published in the Molecules journal, but after it has been improved. I have outlined some suggestions for its improvement.

  1. The first sentence of the abstract should be modified or removed. Definitions of terms can be provided in the introduction, but the abstract should offer an overview of the entire content presented in the article.
  2. Regarding the structured layout of the introduction, fragmented paragraphs can make the text more difficult to follow and assimilate. It might be more beneficial for the reader to have a smoother presentation of the ideas in the introduction, without pronounced divisions. I suggest consolidating the paragraphs to achieve a smoother transition between topics, improving coherence and readability. This will facilitate understanding and absorption of the information presented, ensuring that the flow of ideas is communicated in a more efficient and coherent manner.
  3. Figure 1a has a much lower resolution compared to Figure 1b.
  4. Section 2. General aspects of signal generation mechanisms are detailed, but I believe more examples from the literature (if available) and critical comparisons between them are necessary. For example, in section 2.1, only two studies from the literature addressing FRET are described.
  5. Figure 2 requires a better resolution.
  6. Section 3.1 appears challenging to navigate. The English is not very fluent, and the bullet points do not facilitate smooth reading. Only two aptasensors are mentioned from the literature, with more emphasis on theory. Describing and reformulating the method approached by another researcher is not sufficient, in my opinion. A small comparison highlighting similarities and differences, advantages, and disadvantages is necessary in each section.
  7. Some bibliographic references lack their respective numbers, but this aspect is likely to be resolved later.
  8. In section 4, a table summarizing the described examples along with signal amplification strategies and possibly a specific evaluated parameter would be helpful to provide a clear picture of the existing opportunities.
  9. In the conclusion section, I believe citations should be avoided, and instead, it should contain your own opinions on the topic discussed. It would also be useful to propose an idea or concept as future research perspectives in this field.
  10. For a review that delves into such complexity, 103 references seem relatively few. Adding more concrete examples for each signaling strategy or generation method could be beneficial.
  11. The English and sentence length need improvement.
Comments on the Quality of English Language

In my opinion, the main improvement would consist of providing clear and more numerous examples of the approaches reported in the literature at this time. I believe the article could be successfully published in the Molecules journal, but after it has been improved.

Round 2

Reviewer 1 Report

Comments and Suggestions for Authors

The quality of this review was improved after revision. It can be accepted in current version.

Comments on the Quality of English Language

Minor editing of English language required

Reviewer 2 Report

Comments and Suggestions for Authors

The article has been significantly improved, and I believe it is ready for publication.